# Biological age estimation using circulating blood biomarkers

Jordan Bortz [1,2 ✉], Andrea Guariglia[1,2], Lucija Klaric[1], David Tang [2], Peter Ward[1], Michael Geer[1], Marc Chadeau-Hyam [2,3,4], Dragana Vuckovic [2,4,6 ✉] & Peter K. Joshi [1,5,6 ✉]

Biological age captures physiological deterioration better than chronological age and is amenable to interventions. Blood-based biomarkers have been identified as suitable candidates for biological age estimation. This study aims to improve biological age estimation using machine learning models and a feature-set of 60 circulating biomarkers available from the UK Biobank ($n = 306,116$). We implement an Elastic-Net derived Cox model with 25 selected biomarkers to predict mortality risk (C-Index = 0.778; 95% CI [0.767–0.788]), which outperforms the well-known blood-biomarker based PhenoAge model (C-Index = 0.750; 95% CI [0.739–0.761]), providing a C-Index lift of 0.028 representing an 11% relative increase in predictive value. Importantly, we then show that using common clinical assay panels, with few biomarkers, alongside imputation and the model derived on the full set of biomarkers, does not substantially degrade predictive accuracy from the theoretical maximum achievable for the available biomarkers. Biological age is estimated as the equivalent age within the same-sex population which corresponds to an individual's mortality risk. Values ranged between 20-years younger and 20-years older than individuals' chronological age, exposing the magnitude of ageing signals contained in blood markers. Thus, we demonstrate a practical and cost-efficient method of estimating an improved measure of Biological Age, available to the general population.

[1] Humanity Inc, Humanity, 177 Huntington Ave, Ste 1700, Humanity Inc - 91556, Boston, MA 02115, USA. [2] Department of Epidemiology and Biostatistics, School of Public Health, Faculty of Medicine, Imperial College London, London, UK. [3] MRC Centre for Environment and Health, School of Public Health, Imperial College London, London, UK. [4] NIHR-HPRU, Health Protection Research Unit in Chemical and Radiation Threats and Hazards, Public Health England and Imperial College London, London, UK. [5] Centre for Global Health Research, Usher Institute, University of Edinburgh, Edinburgh, UK. [6] These authors jointly supervised this work: Dragana Vuckovic, Peter K. Joshi. ✉email: jordan.bortz@humanity.email; d.vuckovic@imperial.ac.uk; peter.joshi@humanity.email

Biological age, a concept increasingly discussed since its introduction by Baker and Sprott in 1988[1], challenges the notion that chronological age is always the best predictor of physiology or function. In accordance with Mahdi et al.[2], we define biological age as a latent conceptual value, reflecting the extent of aging-driven biological changes, such as molecular and cellular degradation. We estimate this via the use of its prognostic effect on a strongly age-related outcome: mortality, the ultimate measure of biological and functional decline. Whilst the concept of biological age was introduced over three decades ago, practical difficulties in estimation persist. Discovering biomarkers to assess biological age offers two potential benefits: evaluating the effectiveness of aging interventions and making better predictions of age-related conditions, including mortality, using a scale familiar to laypeople.

In the past few decades, biological age has been estimated using a variety of biomarkers – telomere length, DNA-methylation, proteomics, metabolomics, glycomics, wearable sensor data and blood-based, clinical biomarkers[3–8]. Composite blood-based biomarkers have demonstrated an ability to detect differences in biological age even in cohorts of young and healthy individuals, prior to the development of disease or phenotypic manifestations of accelerated ageing[9]. When contrasted against some of the omics-based biological age estimates, such as telomere-based or epigenetic clocks, blood biomarkers have considerable cost and scalability advantages[10,11]. However, the number of studies on blood biomarker-based biological age estimation remains low and further validation is required[3,12]. By making use of a dataset of unprecedented size, this study aims to improve on biological age estimation using machine learning methods and address real-world drawbacks, such as sparse data and cost.

Machine learning techniques have proved to be popular choices in the construction of biological age estimates[3,12–14]. One of the most relevant studies in this domain is Levine et al.'s[15] 2018 paper, wherein the authors developed a biological age measure called PhenoAge, using a Cox proportional-hazards model with an Elastic-Net penalty on data consisting of forty-two blood-based biomarkers and all-cause mortality collected within the NHANES programme. Liu et al.[16] subsequently demonstrated that PhenoAge was significantly associated with all-cause and cause-specific mortality, even after adjusting for chronological age and sex.

In this study, we used a dataset of 57 blood-biomarkers (Supplementary Table 1) and all-cause mortality from 306,116 participants from the UK Biobank (UKBB) dataset (https://www.ukbiobank.ac.uk/). Participants' ages ranged from 38 to 73 years, with a mean of 56.3 years. The overall mortality rate of the population was 4.3% for females (6502) and 7.8% for males (12,186), with a total of 18,688 recorded deaths. Follow-up duration ranged from 0.01 years to 14 years (average of 11.6 years) (Supplementary Table 2). We used distinct geographies for training and validating our machine-learning models. The training set comprised of participants from England ($n = 269,652$) and Wales ($n = 13,481$), whilst the test set was made up of participants in Scotland only ($n = 22,983$).

We built an Elastic-Net penalised Cox proportional-hazards model, with stability analysis, as the foundation of our biological age prediction, using survival time as outcome and blood biomarkers as predictors, while adjusting for sex and age, which were not penalised. We then convert predicted mortality risk – our proxy for functional age – into biological age using an empirically demonstrated link between mortality hazard ratios and aging. As an alternative we also constructed a Random Survival Forest (RSF)[17]. As pointed out by Qiu et al.[18], who also investigated the ability of blood markers to predict biological age on the UKBB dataset, ensemble tree methods, such as RSFs and gradient-boosted trees, can capture non-linearities and interaction effects between predictors. Predictive performance of all models was evaluated using Harrell's Concordance Index ("C-index" or "concordance statistic"), which is the most used measure of predictive discrimination in the context of survival models[19,20].

Our Cox and RSF models performed similarly, with near-identical C-Index confidence intervals. The two models demonstrated an 11% and 9% relative increase in predictive value, respectively, when compared against PhenoAge[15] when all were applied to the Scottish test set.

We subsequently show that using common clinical assay panels, with few biomarkers, alongside k-Nearest Neighbour imputation and our Elastic-Net model, does not substantially degrade predictive accuracy from the theoretical maximum achievable for the available biomarkers. Our Elastic-net Cox model was then used to estimate biological age values, which took on values between 20 years younger and 20 years older than individuals' chronological age.

## Results

We trained 100 Cox proportional-hazards models with Elastic-Net penalty, each on a 50% subsample of the training data. Using a stability selection approach, we calculated the per-variable selection proportions across each of the 100 calibrated models as a proxy for their importance. We considered variables with selection proportion >80% to be stably selected (Fig. 1a). A total of 25 (of the 57) biomarkers were stably selected, of which 23 had a selection proportion >95%. Reducing the selection proportion threshold to 50% would only have resulted in the inclusion of 4 additional variables. The ensemble of 25 stably selected features along with age and sex were included in a final unpenalised Cox model which we label the "Full Elastic-Net Cox (ENC)" model. The Full ENC model resulted in a C-Index of 0.778 with 95% confidence interval of [0.767–0.788] on the Scottish test set, showing improvement over (i) the model including sex and age only (null model) with a C-Index of 0.726 [0.715–0.737] and (ii) the PhenoAge model with C-Index of 0.750 [0.739–0.761] in the Scottish test set (Fig. 1b).

The tuned RSF yielded a very similar result to the Full ENC model with a C-Index estimate of 0.773 [0.758–0.787] on the test set (Fig. 1b), hence not supporting non-linear effects nor the existence of complex interactions across biomarkers. With a value of 0.5 corresponding to random prediction, we consider the predictive uplifts of models to be the additive increase above 0.5. The additive increases of 0.278 and 0.273 for the Full ENC and RSF models, compared to 0.226 for the null model, indicate a 22.7% and 20.5% increase in predictive value, respectively. Similarly, both models outperformed the PhenoAge model applied on the UKBB data, with a 0.028 and 0.023 increase in C-index (11% and 9% increase in predictive value respectively).

As a sensitivity analysis to assess predictive ability and generalisability of the ENC model we stratified the test set by a self-reported, binary indicator of "long-standing illness, disability, or infirmity", which was available at assessment date and straightforward to use[21,22], and reported the corresponding C-Indices yielded by the null and ENC models trained on the full training dataset (Fig. 1c). The C-Index for those not reporting a prevalent condition at baseline ("Healthy") was 0.761 [0.746–0.776] for the ENC model, which is higher than that from the null model at 0.735 [0.720–0.750]. In the group of participants reporting a prevalent condition ("Sick") the estimated C-Index was comparable at 0.760 [0.744–0.776] for the ENC model and 0.687 [0.671–0.704] for the null model. In both groups of participants, the ENC model yielded an increase in the C-index compared to that of the null model, corresponding to 11% and 39% increases

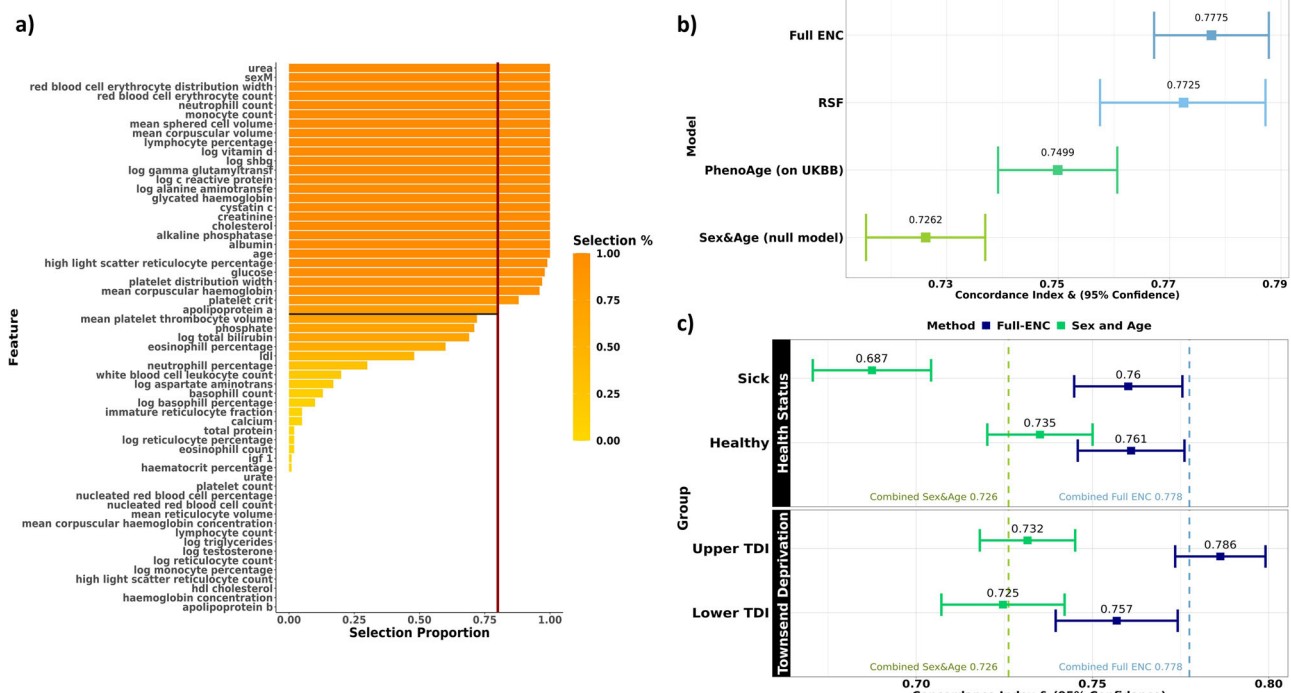

**Fig. 1 The Full Elastic-Net Cox model performs similarly to the Random Survival Forest and produces robust mortality risk predictions across both healthy and sick groups, and across groups of differing socio-economic status. a** Selection proportions of each feature as a percentage of the 100 Elastic-Net iterations performed, ranked from highest to lowest. The adopted selection threshold of 80% is indicated in red. **b** Forest plot comparing the C-Index values (and 95% CI) of (1) a Cox model using sex and age only (null model), (2) the PhenoAge model applied on the Scottish UKBB data, (3) our RSF and (4) our Elastic-Net derived Cox-model (Full ENC). **c** Comparison of C-Index values of the Full ENC and sex-and-age-only null models for (i) Healthy and Sick groups and (ii) for Lower and Higher rated Townsend Deprivation indexed groups. Across both stratifications, C-Index values of the Full ENC were significantly higher than those produced by the null model, with non-overlapping, or near non-overlapping* confidence intervals, indicating that the Full ENC model provides a statistically significant uplift in predictive ability. The dashed vertical lines represent the C-index values of the Full ENC and the null model on the full Scottish test set. *Whilst the separate confidence intervals of the Full ENC model and the null model visually overlap for the "Healthy" and "Lower TDI" groups, the T-test for the contrast shows a significant (p<5%) difference in C-Index values.

in predictive value, for the Healthy and Sick groups respectively. This stratified analysis suggests that the blood biomarkers detected by the ENC model are jointly predictive of mortality risk irrespective of prevalent morbidity.

As a further sensitivity analysis, to investigate the generalisability of the model across groups of varying socio-economic status, we stratified the test set into two groups based on the Townsend Deprivation Index ("TDI")[23] of each participant's postal code, using the 50[th] percentile as the split point. The C-Index for those with an above-average deprivation score ("Higher TDI") was 0.786 [0.773–0.799] for the ENC model; higher than that from the null model at 0.732 [0.718–0.745] (Fig. 1c). In the group of participants with a lower-than-average deprivation score ("Lower TDI") the estimated C-Index was 0.757 [0.740–0.774] for the ENC model and 0.725 [0.707–0.742] for the null model. Again, in both groups, the ENC model demonstrated an increased C-Index against the null model, with a 23% increase in predictive value for the more deprived group, and a 14% increase for the less deprived group. This stratified analysis suggests that the blood biomarkers detected by the ENC model are jointly predictive of mortality risk across the socio-economic spectrum in the UKBB.

Practical application of blood-based biological age estimation will be facilitated by the use of existing results from assays that individuals have obtained for various clinical reasons. In principle, any subset of the 25 biomarkers might be available (i.e. >33 million combinations), although, in reality, clinical practice will tend to result in some subsets being more common than others.

We researched which panels corresponded to the most performed blood tests in the UK as part of regular medical check-ups, employer-based testing, or diagnostic testing, within the NHS or by private providers[24–26] and identified the 10 most commonly available panels of markers (Supplementary Table 3). Similar, bespoke Elastic-Net Cox models were developed on the training set for each of these 10 representative panels of markers separately and we report the corresponding C-Indices in the Scottish test set (Fig. 2, maroon bars). These C-Indices estimate the theoretical maximum of predictive accuracy for individuals who have had only these assays measured.

The performance of each of these per-panel models was compared with the performance of the Full ENC model, where we considered biomarkers not included in the panel to be unavailable, and were thus imputed using the k-Nearest Neighbour (kNN) method[27].

For all panels, there were no significant (p > 5%, T-test for the contrast in means) differences in C-Index values across both methods, with the average reduction in C-index across all panels at 0.008. These results appear to suggest that there is no substantial loss in predictive accuracy when imputing and using the ENC model compared to bespoke models.

After imputation, the ENC model for the top five performing panels (panels 4,8,7,9,10) yielded an average improvement in C-Index of 0.031 over the sex-and-age-only null model (Fig. 2, green line), and an average reduction in C-index of 0.021 against the Full ENC model with all 25 selected biomarkers available (Fig. 2, blue line).

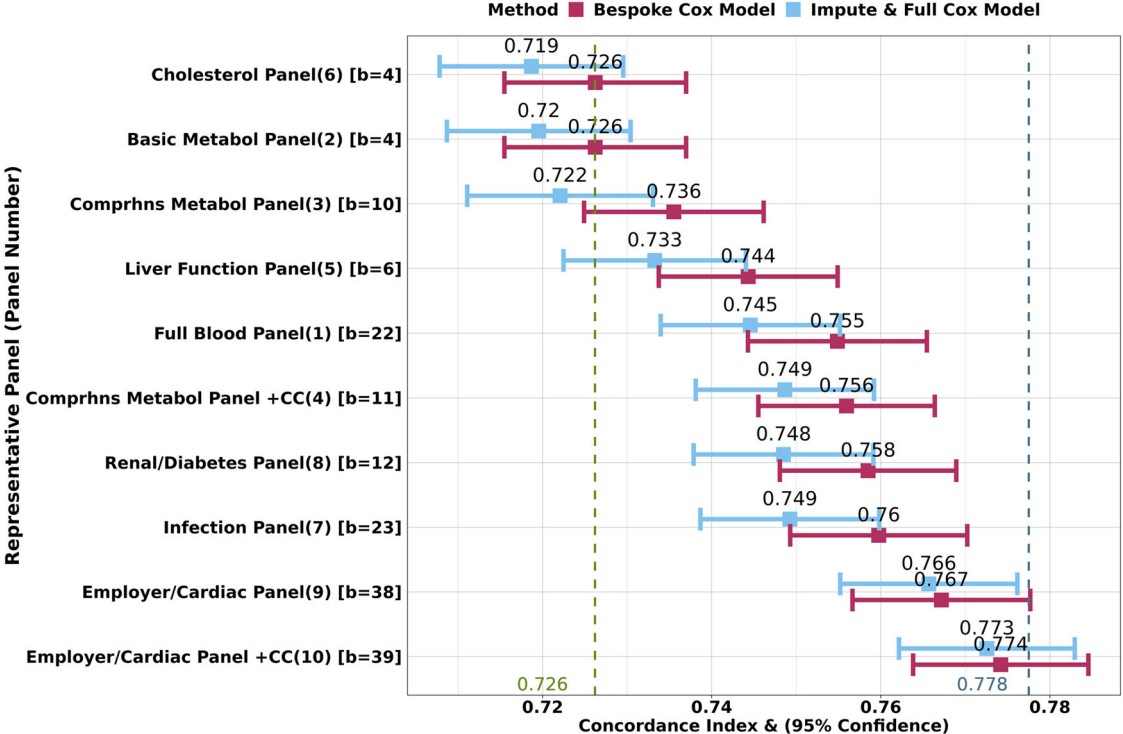

**Fig. 2 Imputing out-of-panel biomarkers and using the Full ENC model did not substantially reduce predictive accuracy compared to bespoke models for each representative panel.** Comparison of concordance values across bespoke models vs imputed ENC models, for each of the 10 real-world representative blood panels, on the Scottish test set. Sex and age were also included as (unpenalised) features in all models. The number in brackets next to the panel name indicates the panel number as per Supplementary Table 3. The number in square brackets indicates the count of biomarkers (b=) in the panel. The performance of the impute-then-Full-ENC method is similar to that of the bespoke models, especially for the more comprehensive panels. The green dashed vertical line indicates the 0.726 C-Index of the (sex and chronological age only) null model, whilst the blue line indicates the 0.778 C-Index of the Full ENC model with all 25 selected biomarkers measured. "+CC" indicates the addition of cystatin C to the panel (panels 4 and 10).

Standardised model coefficients and 95-percent confidence intervals on the log hazard scale for the Full ENC model are displayed in Fig. 3a. Increasing age, male sex and increased levels of cystatin C appear to have the strongest effects towards an increasing hazard rate, whilst high levels of creatinine, alanine aminotransferase and vitamin-D are associated with lower hazard rates. For the RSF, variable importance (VIMP) values were also generated. As was seen in the Full ENC model, age is the most important variable, followed by cystatin C, which appears to have a higher importance value than sex in the RSF (Supplementary Fig. 2). Whilst Chan et al.[28] did not focus exclusively on blood markers, their use of principle components analysis and Klemera and Doubal's method[29] of constructing biological age also identified cystatin C as a feature of importance (second most important for females and fourth most important for males) in their study of biological age modelling in a healthy subset of the UKBB. Similarly, Qiu et al.'s[18] gradient boosted tree method indicated that cystatin C was the feature of primary importance in the prediction of all-cause mortality for UKBB females over the age of 65.

The ENC model suggests that a one standard deviation increase (i) in age (8.1 years) increases mortality hazard by 84% [95% CI: 80% - 88%], (ii) in cystatin C (0.14 mg/L) increases mortality hazard by 31% [28% - 33%], when adjusted for sex and all selected biomarkers (Supplementary Table 4). Our result mirrors the finding of a meta-analysis by Luo et al.[30], which identified a 32% [12% - 55%] increase in all-cause mortality hazard with a one standard deviation increase in cystatin C, based on 39,000 participants across nine different studies. The addition of cystatin C to the Comprehensive Metabolic Panel and the

Employer/Cardiac Panel confirmed the incremental predictive uplifts in models where this biomarker is present, increasing the C-index by 0.02 and 0.007 for each of the Panels respectively (Fig. 2, panels 3 vs 4 and 9 vs 10). Red blood cell (erythrocyte) distribution width appears to have the strongest effect size after cystatin C, with a standard deviation increase implying an 18% [16%–20%] increase in mortality hazard.

The Full ENC model coefficients are compared to the coefficients as per Levine et al.'s PhenoAge[15] model in Fig. 3b. The 10 variables in the PhenoAge model, derived from the NHANES dataset, were also stably selected in our ENC model except for overall white blood cell(WBC) count, which was substituted in our model by individual WBC components (monocytes, neutrophils, and lymphocytes). Model coefficients appear to be remarkably similar between the PhenoAge model and our ENC model, except creatinine, which had a small, negative effect size in our model, in contrast to a small, positive effect in the PhenoAge model. Overall, these consistent results provide validation of the effect that these variables in two large, independent cohorts. Our model additionally included sex and selected 17 additional biomarkers, some of which have the largest standardised coefficients and were not considered in the derivation of PhenoAge. The complete set of model coefficients is also shown in Supplementary Table 4 alongside implied hazard ratios.

Due to the documented positive relationship between serum cystatin C and creatinine[31,32], as well as the positive correlation ($r = 0.51$) identified in our descriptive analysis (Supplementary Fig. 1), we investigated how the effect sizes of these two biomarkers were altered with the removal of the other. When creatinine was removed, the standardised effect size for cystatin C

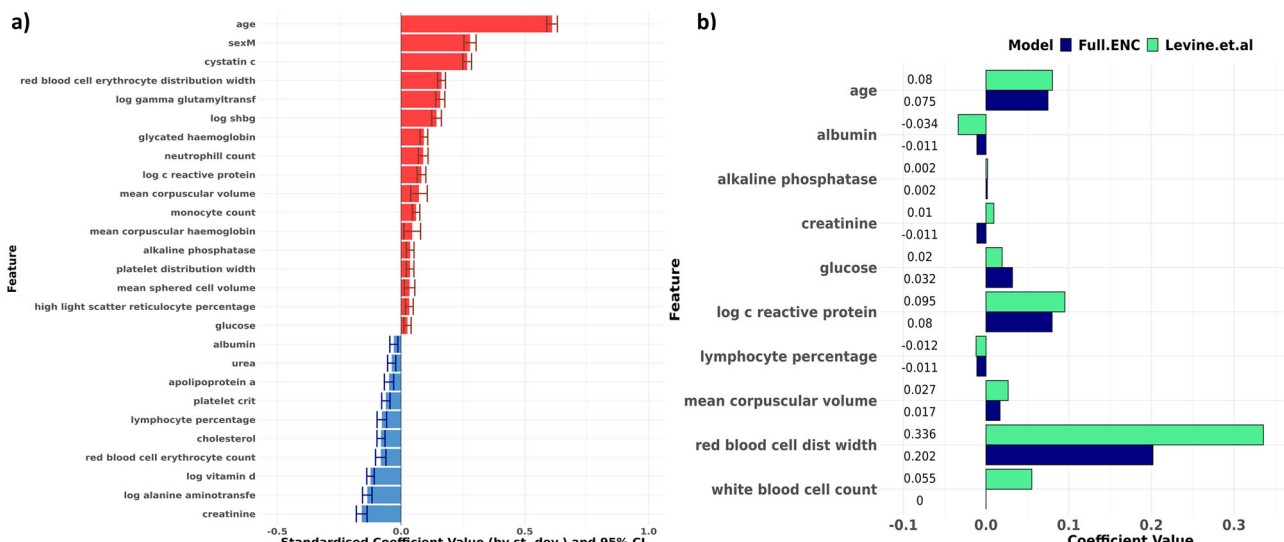

**Fig. 3 Our results confirm relationships suggested by Levine et al.'s PhenoAge model, and additionally suggest that cystatin C is the biomarker of primary importance in biological age estimation. a** Bar chart showing standardised Cox model coefficients and 95% confidence intervals (log hazard scale) of the Full ENC model developed using stably selected variables, ranked in descending order. Coefficients are standardised (i.e. rescaled) by multiplying by the standard deviation of the variable concerned. Red indicates that higher levels increase mortality hazard; blue indicates that higher levels reduce mortality hazard. Apart from age and sex, cystatin C appears to have the strongest effect size. **b** Comparison of coefficient values between Levine et. al's PhenoAge coefficients (green) and our Elastic-Net derived Cox model (blue). Model coefficients are similar across both models. Our ENC model selected individual WBC components (monocytes, neutrophils and lymphocytes) rather than overall WBC count. Measurement units for biomarkers were the same across both models.

was still the largest of all blood biomarkers. The hazard ratio decreased to a 22% [20%–24%] increase in mortality hazard for one standard deviation increase in cystatin C level. When cystatin C was removed, the effect size for creatinine became slightly positive, but was no longer statistically significant ($p > 5\%$, T-test against null hypothesis of 0), with a one standard deviation increase implying a 0% [−2% to 2%] reduction in mortality hazard.

The ENC model was used to estimate Biological Age Acceleration (BAA), representing the additional years of physiological deterioration above that implied by one's chronological age (i.e., the difference between biological age and chronological age), as explained in Methods. Most BAA values ranged from −20 to 20 years, with an interquartile range of 7 years, centered around a median value of –0.61 with a slight right skew (Fig. 4a), similar to the results obtained by Liu et al.[16]. However, the range of age acceleration seen in Liu et al.[16], of −3 to 6, is substantially narrower than our BAA range, which could at least partially be attributed to the differences in methodology used to derive BAA. It could also represent the additional sensitivity introduced in our model due to the inclusion of 17 additional circulating biomarkers, as well as the adjustment for sex.

As proposed by Liu et al.[16], we considered 3 broad age groups ([≤50], (50–65], (65+]), and plotted the Kaplan–Meier curves for the top and bottom quintiles of BAA values within each age group (Fig. 4b–d). The plots demonstrate that BAA provides an indication of biological age and mortality risk even at the youngest ages in the dataset (<50 years of age at recruitment). The log-rank test was significant for all age groups ($p < 0.0001$). Notably, the mortality rates for those in the highest BAA quintile in an age group were higher than those in the lowest BAA quintile in the older age group. For example, those in the highest quintile of BAA in the <=50 age group experienced higher mortality rates than those in the lowest quintile of the 50–65 age group, over the next 14 years of follow-up.

As a final sensitivity analysis, we conducted similar analyses in males and females separately (the details of which are discussed in the Supplementary Note, with results illustrated in Supplementary Figs. 3–7 and Supplementary Table 8). The resulting C-index values, effect size estimates, and BAA predictions were similar to those obtained in the full population. Sex-stratified models did not produce more accurate predictions than the Full ENC model, even within same sex groups.

## Discussion

In this analysis, we demonstrated that circulating biomarkers have the potential to form the foundations of an accurate and low-cost measure of biological age, via a simple formula. Our biological age estimates ranged between 20 years younger and 20 years older than individuals' chronological age, exposing the extent of ageing signals contained within the selected biomarkers. Our Elastic-Net derived Full ENC model, which selected 25 biomarkers, outperformed the alternative blood-biomarker-based PhenoAge model[15], providing an 11% relative increase in predictive value. The improved model performance likely arises from the large volume of data used for training, and, more importantly, the additional biomarkers cystatin C and red blood cell distribution width being available.

Importantly, the model developed in this study possesses translational value in the real-world setting, where pre-existing measures of blood biochemistry will often be available but will vary greatly across individuals. We established that imputing values for unmeasured blood markers and subsequently using the Full ENC model, did not substantially degrade predictive accuracy away from what would be achieved by developing bespoke, per-panel models. The average reduction in C-Index across all panels, including the poor-performing panels, was 0.008. This suggests that the Full ENC model could be used to estimate biological age values practically and accurately, in a real-world setting, irrespective of the number of biomarkers available,

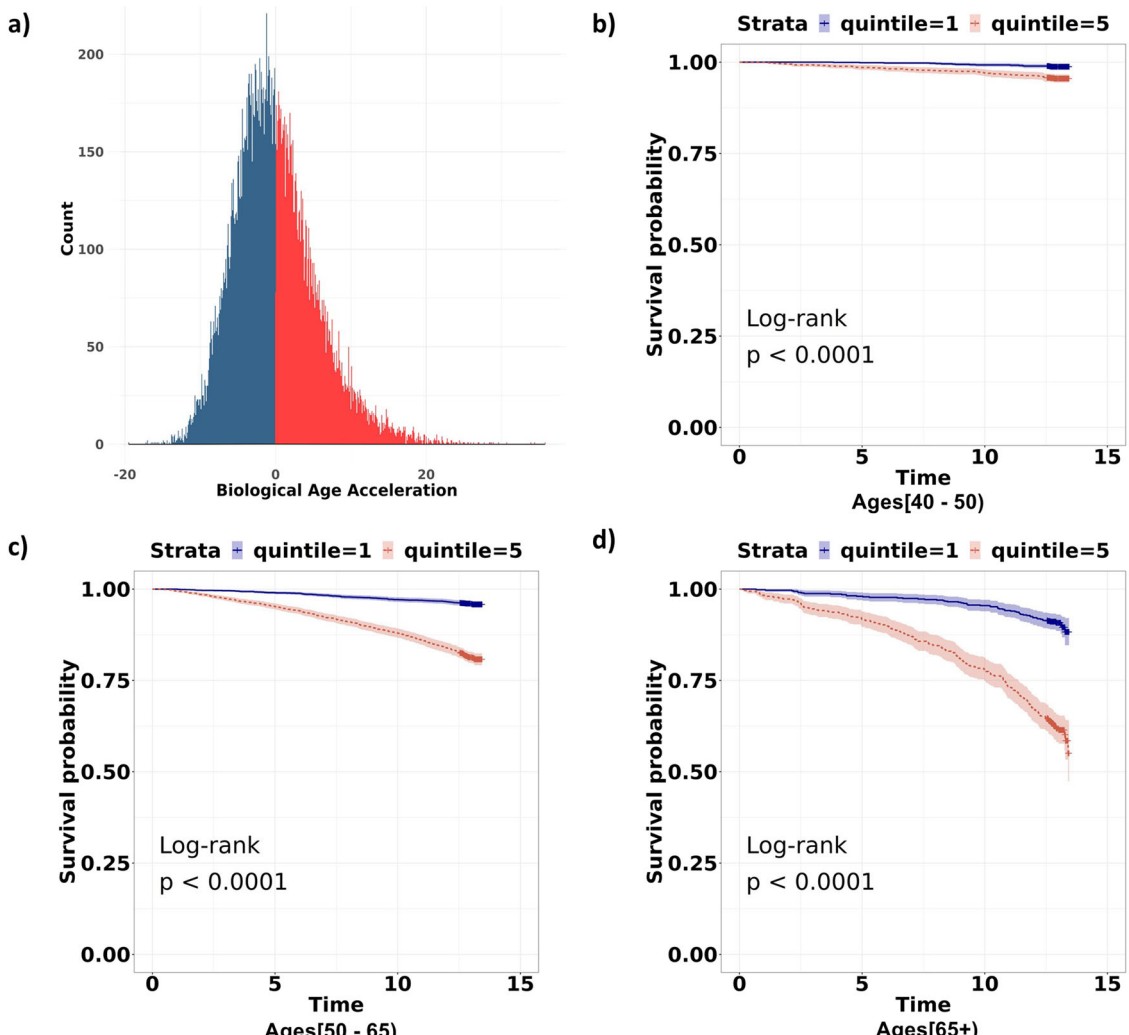

**Fig. 4 Biological Age Acceleration values range between −20 and 20, and reflect mortality risk even in same-age groups. a** Distribution of estimated BAA values on the test set. Blue indicates a negative BAA (biological age < chronological age), and red indicates a positive BAA (biological age > chronological age). The distribution is largely symmetric around 0, with most values ranging between −20 and 20. **b–d** Kaplan-Meier curves comparing survival probabilities for the top and bottom BAA quintiles for each of the three age categories in the test set. Blue indicates the bottom quintile (largest negative BAA values) whilst red indicates the top quintile (largest positive BAA values). Notable differences are observed between quintiles, especially at older ages.

provided that an existing dataset is available with at least the full panel of 25 blood markers (for example UK Biobank, or similar). This approach is more computationally feasible than building a new bespoke Cox model for the available assays from scratch using more than 300,000 UK Biobank subjects. The ability of our BAA to distinguish between high risk and low risk individuals within same-age and same-health groups in our sample is an indication that BAA can identify physiological deterioration even in younger and healthy populations.

Our findings follow the trajectory of the growing literature in the field of biological age, where numerous clocks have been developed that are predictive of future morbidity and mortality outcomes[3,6]. In particular, our focus on practicality and translational value was influenced by the blood-based PhenoAge[15] and GrimAge[33] biological age estimators, which were shown to correlate with all-cause mortality even in same-age groups. We have demonstrated that such ageing clocks can indeed be employed highly cost effectively, using pre-existing or readily available blood results. We recognise that our clock could be enhanced by the inclusion of other readily available or easily obtainable health

measures, such as blood pressure, grip strength, and forced expiratory volume, but leave this extension of scope to future work.

In the Full ENC model creatinine had a small, negative effect size, in contrast to a small, positive effect in the PhenoAge[15] model. However, we do not suggest that increased creatinine is a marker of good health (whilst Levine et al.[15] suggested it marks poor health), and, in fact, creatinine's coefficient was not statistically significant when cystatin C was not present. The contradiction of results against Levine et al.[15] is explained by the presence of cystatin C in our panel, which covaries with creatinine. We believe our results suggest that increased creatinine in the presence of unchanged cystatin C may be desirable. In practice, increased creatinine will often be accompanied by increased cystatin C and a predicted worsening in health outlook.

Our study benefited from several important strengths including the large sample size and the substantial number of biomarker measurements available, which enabled the development of robust inference, including stratified analyses and the use of interpretable machine learning techniques. However, the use of

the UKBB dataset comes with well-documented limitations. The healthy volunteer bias inherent in the selection of participants, as well as the ethnic composition of participants (95% white), have been well documented in the literature[34]. Our analysis of sub-groups stratified by socio-economic status does, however, indicate that our findings are consistent across both more and less deprived groups and thus likely to be similar in a fully representative sample. The homogeneity of the study population as well as the standardisation of measurement protocols across blood samples within the UKBB may not reflect systematic variation in blood results across different laboratories and populations. We do, however, acknowledge the remarkably similar parameters of Levine et al.'s[15] PhenoAge model, which were based on the North-American, NHANES dataset, providing an independent validation of our findings using an external data source. An additional limitation relates to our imputation approach, which occurred within the single UKBB cohort. This approach may overestimate imputation quality in reality, where relationships between biomarkers in different population groups may differ from those observed in the UKBB.

Whilst our training and validation both took place within the UKBB dataset, we have demonstrated that we also have predictive power in a distinct (if adjacent) geographical region. We believe our findings will generalise beyond the UK, and further research - gathering cystatin C in particular – is required to do so. At the same time, we must acknowledge that whilst our predictor did improve on the PhenoAge algorithm in Scotland, we did set PhenoAge a harder challenge: to predict mortality risk in a cohort geographically distant from the training cohort.

Our findings are consequential in an era of rapidly ageing populations. Practical mechanisms for identifying biological ageing and scientifically informed ageing-reversal interventions are essential to maximising population healthspan and reducing pressure on healthcare providers.

## Methods

**Data source and data processing.** The data used in this study forms part of the UK Biobank (UKBB) dataset. The 502,536 participants in the UKBB, recruited between 2006 and 2010 in England, Scotland, and Wales, were aged between 37 and 73 years at recruitment[35]. Participants represented a range of socio-economic, ethnic, and urban/rural population groups[35]. Full details of the UKBB study, including details of recruitment and data collection, can be found on the UKBB website (https://www.ukbiobank.ac.uk/), and supplementary information can be obtained in Sudlow et al.[35]. The specific data fields used in this analysis were date of recruitment, age at recruitment, sex, date of death (if applicable), presence of illness or disease at recruitment (self-reported), the Townsend Deprivation Index recorded for each participant's postal code and a range of 60 blood biochemistry and haematology markers, samples of which were obtained from participants at recruitment. The full list of the blood biomarkers available in the UKBB, prior to the data cleaning and further sub-selection detailed below, can be found in Supplementary Table 1, alongside their measurement units. Additional details on these markers can be found in the UK Biobank Companion Documents[36–38].

Data fields were examined for missing values and any fields where more than 20% of observations contained missing values were removed (lipoprotein-A, oestradiol and rheumatoid factor). For the remaining fields, the underlying reasons for missingness were investigated, per biomarker, using the UKBB data showcase resources (https://biobank.ndph.ox.ac.uk/showcase/label.cgi?id=100080). Blood assay data in UKBB may be missing for a variety of reasons, such as aliquot or aliquot dilution problems, values

being below the reportable limit, and no data being obtained from a sample. An examination of the underlying causes informed subsequent handling of data, and the appropriateness of proceeding under a complete case analysis. For example, if missing values were largely driven by measured levels being below the minimum limit of detection, omitting such individuals would exclude individuals with low levels of a biomarker from this study, biasing results. For all biomarkers, apart from testosterone, the reasons for missing values appeared to be independent of both the biomarker and the participant and were primarily caused by aliquot errors or aliquot dilution problems. We therefore proceeded under a complete case analysis, assuming that data was largely missing completely at random. The final dataset consisted of 306,756 observations, each having 57 blood biomarkers, alongside sex and age. Data was examined for outliers, and a log-transformation was applied to variables with strong positive skew (alanine aminotransferase, aspartate aminotransferase, gamma glutamyltransferase, C-reactive protein, sex hormone binding globulin (SHBG), testosterone, total bilirubin, triglyceride and vitamin-D). The final dataset was then split into a training set (participants based in England and Wales) and test set (participants based in Scotland), and these were used in the remainder of the study.

**Model Performance.** Predictive performance of all models was evaluated using Harrell's Concordance Index ("C-index" or "concordance statistic"), which is the most commonly used measure of predictive discrimination in the context of survival models[19,20]. Concordance is defined as the probability that the magnitude of the predicted values for observations i and j (pi and pj), rank in the same order as the actual values for these observations, ai and aj:

$$P(pi > pj | ai > aj).$$

In other words, the probability that, for a pair of observations, the observation that experiences an event first, had a worse predicted outcome. The pair of observations is considered concordant if the rankings of the predicted and actual values are the same and are discordant otherwise. Ignoring ties, the Concordance Index is then defined as:

$$ConcordantPairs / (ConcordantPairs + DiscordantPairs)$$

A concordance value of 0.5 corresponds to a model that is no better than a random guess, and a value of 1 would imply a perfect model. In the presence of censoring, some pairs cannot be compared or classified as concordant nor discordant. For example, pairing an individual who is censored at year 5 against an individual who experiences a mortality event at year 10[20].

Our models were compared against the performance of the well-known PhenoAge model. Kuo et al.[39] recently highlighted differing genetic dimensions of aging clocks, comparing PhenoAge and the alternative BioAge[40]. We restricted the scope of our comparative analyses to the former as it was trained using blood biomarkers as predictors and mortality as outcome, as with our approach.

**Cox proportional-hazards model with elastic-net penalty and stability analysis.** Using the training dataset, we developed 100 Elastic-Net Cox proportional-hazards models, using survival time and mortality indicator as outcome, each using a subsample of 50% of the dataset, as suggested by Bodinier et al.[41]. For each iteration, 10-fold cross validation was used to select the optimal parameters, lambda and alpha, that minimised the partial likelihood deviance statistic, across a grid of 80 lambda values and alpha values between 0 and 1. As is commonly done in penalised

regression models, we selected the (larger) lambda value that corresponded to a deviance statistic of one standard error above the minimum during cross-validation (lambda 1-s.e.)[42]. The penalisation was only applied to the biomarker variables and neither sex nor age were penalised in any of the Elastic-Net models. All variables were standardised for the Elastic-Net optimisation process.

Stability analysis was performed to improve the predictive accuracy and the reliability of the model, ensuring that variable selection was not driven by outliers or a particular subset of observations[41]. For each variable, we calculated the proportion of the 100 iterations in which the variable was selected, and its standardised coefficient value, as indicators of its relative importance. A selection proportion of 80% was used to determine if a variable was selected stably[43]. These stably selected variables were then included in a non-penalised Cox model, to produce the final regression equation, which was used to generate predictions. This model is labelled the Full Elastic-Net Cox (ENC) model, where "full" indicates the use of all available biomarkers in model training along with age and sex.

This approach was used both for the model developed on the full set of biomarkers as well as for the 10 bespoke models developed for each of the 10 representative real-world blood panels. The Elastic-Net models were developed using the *survival*[44] and *glmnet*[45] packages in R version 4.1.3[46].

**Biological age acceleration**. We required an interpretable format of presenting an individual's relative increase or decrease in mortality risk in relation to one's chronological age, based on blood biochemistry. Biological Age Acceleration (BAA) is defined as the additive difference between one's model-implied biological age and one's chromological age, adjusted for sex. To quantify this age acceleration, we decomposed the Cox model equation on the log scale as follows:

$$\log(HazardRate) = \log(BaselineRate) + X_A\beta_A + X_S\beta_S + X_B\beta_B \tag{1}$$

where $X_A$ and $X_S$ each represent the [n x 1] matrix of age and sex covariate values, $\beta_A$ and $\beta_S$ represent the corresponding age and sex effect sizes (beta coefficients). We reformat this equation as:

$$\log(HazardRate) = \log(BaselineRate) + X_A\beta_{A0} + X_S\beta_S + X_{BA}\beta_{BA'} \tag{2}$$

where $\beta_{A0}$ is now the age effect set to the null model effect size, ensuring that the average effect of chronological age on biological age at each age is retained within this chronological age term (rather than reflected in the biomarker component due to covariance). $X_{BA}$ represents the [n × 58] matrix of biomarker values and age, $\beta_{BA'}$ represents the [58 × 1] vector of corresponding biomarker effect sizes and the adjustment to restore the overall effect of age as per Eq. (1) on the log hazard rate. Setting the first 3 terms on the righthand side of the equation as constant K, for an individual of a given age and sex:

$$\log(HazardRate) = K + \log(HazardRatioBloodBiomarkers) \tag{3}$$

where *log(HazardRatioBloodBiomarkers)* is equivalent to the product $X_{BA}\beta_{BA'}$. Using the relationship between log hazard ratio and age as suggested by Joshi et al.[47], where a log hazard ratio of Y indicates an additive change in age of approximately 10Y, we multiplied the product $X_{BA}\beta_{BA'}$ by 10, to quantify the vector of BAA values. That is:

$$BAA = 10 \times X_{BA}\beta_{BA'} \tag{4}$$

We independently checked this log-hazard-ratio:years-of-age relationship, by producing a Kaplan-Meier curve over age using

the full UKBB dataset of 500,000 participants, and separately using data from the English Life Table No.17 (ELT-17)[48]. Based on the results, we were satisfied that the relationship was sufficiently accurate to form the foundations of our BAA estimate. The results from this investigation can be seen in Supplementary Tables 5–7. Note that for the purposes of the BAA calculation, all variables in $X_{BA}$ were centred, by subtracting the (Scottish test set) mean value for each variable in the test set. This implies that BAA would equal to zero for a theoretical individual, of either sex, who holds the average value for each of the model-selected features.

Once BAA estimates had been derived as per the above method for the test set, we examined the range and distribution of BAA across observations. We also examined the association between BAA and mortality, by producing Kaplan-Meier curves for the lowest quintile and the highest quintile of BAA respectively. We did this across three age groups separately: 50 and younger, 50–65, and 65 years and older, which allowed for a comparison of mortality between BAA quintiles across different age groups. Kaplan-Meier curves were constructed using the *survival*[44] and *survminer*[49] packages in R.

**Random Survival Forest**. A Random Survival Forest (RSF) was trained and the resulting C-Index value on the test set was compared to that achieved by the Full ENC model. A RSF is an extension of Breiman's Random Forest method[50], applied to right-censored survival data[17]. In the survival model setting, nodes are split according to the maximisation of the log-rank test statistic. The RSF has the potential to outperform the Cox model, as it can inherently handle nonlinear and interaction effects. We used the *randomForestSRC*[51] package in R to develop our RSF models.

Tuning was iteratively performed, maximising out-of-bag Concordance Index, for three hyper-parameters in particular: number of variables considered at each split, minimum node size and number of trees. The initial tuning was performed on a wide grid of these three hyper-parameter values, with each of the two successive rounds of tuning taking place over a smaller range of values, around the current optimal point. Three possible split points were considered for each continuous variable, at each split. The final optimal values for the number of variables considered at each split, and the minimum node size were 21 and 130 respectively, for an ensemble of 150 trees. The tuned model was then applied to the test set. The predicted outcome, per terminal node, is the Nelson-Aalen estimator for the cumulative hazard function, calculated on all the training observations in that terminal node. The average predicted cumulative hazard function value across all trees is then compared to the actual survival outcome, per pair of observations in the test set, to compute the C-Index.

We also produced Variable Importance (VIMP) scores, which provide an indication of which variables in the dataset hold the highest predictive ability. The VIMP ordering from the RSF was compared against the selection proportions and standardised coefficients from the Cox Elastic-Net stability analysis.

**Representative, real world blood panels**. We researched which panels corresponded to the most commonly performed blood tests in the UK as part of regular medical check-ups, employer-based testing, or diagnostic testing, within the NHS or by private providers[24–26]. These panels are summarised in Supplementary Table 3. We decided to explore certain panels with and without the inclusion of cystatin C. This was done for two reasons. Firstly, cystatin C does not consistently form part of these blood panels; certain sources included it, whilst others did not. Secondly, during

the course of our investigation, we observed the strong predictive ability of cystatin C in our own preliminary results. In light of these two reasons, we decided to test the sensitivity of performance between panels that did and did not include cystatin C.

We investigated the performance of the Full ENC model, applied on each of these representative panels. To do so, for each of the ten panels, we imputed the notionally unmeasured values of the biomarkers in the test set which were required by the Full ENC model, but were not present in the notional panel set, using k-Nearest Neighbours (kNN) imputation (k = 5) based on a random sample of 10,000 individuals from the training set. Variables were standardised for the kNN imputation, which was performed in R using the *VIM* package[27]. For ease of reference, we label this approach the "impute-then-Full-ENC" method. In each of these 10 cases, the performance of the impute-then-Full-ENC prediction was compared against performance of each of 10 bespoke ENC models developed specifically on the biomarkers present in each of the 10 representative panels, alongside sex and age. The bespoke models represent an estimate of the theoretical maximum of predictive accuracy for individuals who have had only these assays measured.

Note that a single, consistent test set was used to evaluate the performance of each and every one of the models in this study, removing any variability that would be introduced by altering the composition of the observations in the test set.

**Reporting summary**. Further information on research design is available in the Nature Portfolio Reporting Summary linked to this article.

## Data availability

The data used for this study is available to all qualified researchers via the UK Biobank data access process. With the exception of individual participant data, the underlying data for the graphs and charts in the figures is available as Supplementary Data and any remaining information can be obtained from the corresponding author upon reasonable request.

## Code availability

The code used and resultant models for this project are the intellectual property of Humanity Inc. Model coefficients and R code demonstrating the application of the model can be found in this GitHub repository, which has been archived in Zenodo[52]: https://github.com/bortzjd/bloodmarker_BA_estimation.

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

## Acknowledgements

We thank the UK Biobank Resource, approved under application 69634. We acknowledge funding from Humanity Inc, a company dedicated to measuring and improving biological age. J.B. was supported by a Commonwealth Scholarship at Imperial College London, funded by the UK Foreign and Commonwealth & Development Office (FCDO). D.V. and M.C.-H. are members of the Health Protection Research Unit in Chemical and Radiation Threats and Hazards, a partnership between Public Health England and Imperial College London which is funded by the National Institute for Health Research (NIHR). Neither the FCDO nor the NIHR had a role in the study design, data analysis, preparation of manuscript, or decision to publish.

## Author contributions

J.B., P.K.J., D.V., D.T., A.G., P.W. and M.G. participated in conceiving the study objectives and design. J.B., P.K.J. and A.G. carried out initial data preparation. J.B. performed the data analysis and model development with support from P.K.J., D.V., L.K. and D.T.; J.B. wrote the first draft of the manuscript. All authors revised and provided critical commentary on the manuscript. M.C.-H. and D.V. provided funding for the project.

## Competing interests

The authors declare the following competing interests: during preparation of this manuscript, P.K.J. and J.B. were paid consultants to Humanity Inc, a company focussed on measuring and developing interventions for Biological Age. L.K. was an employee of Humanity Inc. A.G. was formerly a paid consultant of Humanity Inc. M.G. and P.W. are founders of Humanity Inc and are employees and hold ordinary shares. P.K.J., M.G. and P.W. are partly remunerated under a Humanity Inc share option scheme. P.K.J. is founder of Geromica, a consultancy providing advice on measurement of health and aging. M.C.-H. holds shares in the O-SMOSE company and has no conflict of interest to disclose. Consulting activities conducted by the company are independent of the present work. All other authors declare no competing interests.

## Ethical approval

This work used existing datasets (UK Biobank), for which ethical approval (including informed consent) had been obtained by UK Biobank for health investigation at the time of collection.
