## [Peer Review File · Communications Biology]

Reviewers' comments:

Reviewer #1 (Remarks to the Author):

This article introduces a novel algorithm to quantify biological age based on blood chemistry analytes. The algorithm is developed and validated in the UK Biobank data. Quantification of biological aging from routinely collected clinical parameters is a potentially important frontier in the integration of geroscience and aging biology into public health and medicine. The very large UK Biobank database is a powerful resource for data mining and algorithm development applications. Therefore, this paper has potential for impact. However, the manuscript lacks a thoughtful consideration of what biological age is and the measurement approach it pursues is therefore somewhat challenging to understand. Moreover, the claims made don't match the level of validation undertaken. This is important work. But the paper needs more. Suggestions are below.

BA should be defined the first time it is introduced in the introduction. The initialization needs to be spelled out. But also the authors need to explain what they think biological aging means and what a measure of it should represent. Then then they need to explain why they think multivariate hazard regression of mortality on blood chemistry parameters is a good way to develop a measurement of that construct.

The limitations of the UKB setting should be articulated – at least in the discussion, but perhaps also up front. The PhenoAge to which they compare their measure was derived in a representative sample of the US adult population. UKB is not that. How might the restricted age range, high-SES skew, and other features of the UKB sample affect BA algorithm development?

It appears that all validation testing was conducted using the same UKB data with which the algorithm was trained. This is not sufficient to demonstrate anything other than the potential value of an algorithm and should not be used as the basis for any comparison to PhenoAge.

PhenoAge was developed in the NHANES III sample and validated in the Liu et al. article in fully-independent data from NHANES IV. At minimum, the new algorithm should be trained and tested in distinct subsets of the UKB sample. But would be much better to test the algorithm in a fully independent dataset.

In order to compare performance of this new measure to PhenoAge, the authors should perform their analysis in an independent dataset (i.e. one not used to train either measure). It is not clear if this was done from the paper. The NHANES IV data analyzed by Liu et al. are publicly available and would provide one setting for this, although the difference in nation setting could bias results toward PhenoAge.

The characterization of the improvement relative to PhenoAge should be specified both in relative % increase and absolute percentage point increase terms. 10% better is fine. But this reflects a 2-percentage-point improvement in discrimination. In the abstract and introduction, the authors should at least report the C-statistic for their new predictor. (it's actually quite striking that an order-of-magnitude + increase in sample size for training yields such modest improvement in the C-statistic: 0.74 for PhenoAge vs. 0.76 for this model)

Reviewer #2 (Remarks to the Author):

The authors present a version of BA calculation relying on clinical labs from the UKB dataset, and use it to predict mortality risk. The authors further show that a more condensed clinical assay panel imputed using k-Nearest Neighbor imputation and followed by BA prediction by Elastic-Net model has

comparable performance. The claimed novelty of the study is dealing with real-world data, with all its caveats.

This is a well written paper, with a solid analysis, but I do have several methodological concerns, as well as concerns regarding novelty and generalizability, outlined below.

Major:

- * The target outcome is not age but mortality, but the authors use the term Biological Age. Even for the Biological Age Acceleration (BAA), this index indicates the difference in mortality (not age) between CAge-based information vs. blood-based information (which is not necessary equivalent to the general BAge, the age estimate based on biological information). This terminology should be clarified. It is parallel to PhenoAge, but not to other studies that predict age itself based on omics (see below).

- * The meaning of comparison with "Impute & Full Cox Model" (in Fig. 2) is unclear. For the analysis purpose, the authors first set the variables that is not included in a target panel to NAs for all people in the testing set (regardless these values were actually detected in the original data), and then imputed these NAs based on the detected real values of the people in the training set. Practically (as the authors claimed), this imputation procedure is impossible. So, I don't understand why they presented "Impute & Full Cox Model" in Fig. 2. In other words, the comparisons between the Full ENC model (blue dashed line) and each "Bespoke Cox Model" (red bars) seem sufficient to claim their conclusion. If this is a misunderstanding, the authors should clarify.

- * From the health-stratified analysis (Fig. 1c), the authors concluded that their Full ENC model work similarly for both Healthy and Sick (lines 130–131). However, the reason why they chose to focus on self-reported condition is not clearly mentioned. Based on Fig. 3a, age was the largest predictor for mortality. So, rather than (or in addition to) health status, the authors could perform the stratified analysis with age. In this age-stratified analysis, the model may not work similarly across age-stratified groups, but it would be a rather interesting point; e.g., they could check cystatin c and creatinine while considering their analysis about them (lines 225–231).

- * Similar studies have been performed on the same db albeit with slightly different algorithms/outcomes, e.g. <https://pubmed.ncbi.nlm.nih.gov/33693684/> or <https://www.medrxiv.org/content/10.1101/2022.10.05.22280735v1>, in addition to Levine et al. The authors should better stress their novelty beyond a slight improvement in performance of the model itself.

- * The authors claim that their BA models demonstrated a 9.2% relative increase in predictive value when compared against PhenoAge. The authors should discuss what component of their methods causes this (modest) increase in performance.

- * The ENC model was both trained and evaluated on UKBB data, as the authors acknowledge. To improve the generalizability of their results and substantiate their claim for improved performance, the authors should apply it for the NHANES dataset as an external validation of not only the model performance itself, but also the downstream analysis of feature inference.

Minor:

- * Sex stratified BA models were shown to have significantly different predictive features (e.g. Earls et. al. The Journals of Gerontology: Series A, 2019). The authors do perform stratified model, but limit the discussion to overall performance, rather than focusing on the sex-specific features of both the BA model and the delta age analysis.

- * Some key papers are not cited (mentioned above)

Reviewer #3 (Remarks to the Author):

Bortz and colleagues present a study which builds on previous efforts to estimate biological age using blood-based biomarkers. The main novelty over the main comparison paper (Levine 2018 "PhenoAge"

which used similar statistical methods) is the use of UK Biobank, which is substantially larger than the previous efforts (in terms of sample size, plus number of biomarkers used). I find the paper to be well-written, the analysis thorough, and the results interesting. In particular, I am glad to not see another “biological age” paper where the model is trained to predict chronological age! Instead the authors use the far more interesting and relevant phenotype for prediction/intervention: “mortality.”

Overall, I have no concerns, however it is disappointing that the study authors (i.e., “Humanity Inc.”) have not released the BA model weights despite drawing heavily from previously published (and openly accessible) efforts (especially the Levine 2018 paper, which uses similar methods and this work extends to UKB), meaning their results cannot be used by others. This massively limits the impact and general interest in the work.

My scientific comments are below:

1. Though telomeres are mentioned in sentence 1 as previously used BA markers they are not included in this study. Why not? I could see the argument that it is not commonly measured etc, so please justify the choice of 57 markers more clearly, and the exclusion of some known ageing biomarkers e.g., telomere length. Similar question for blood pressure: why not include? I know not a “blood” biomarker per se, but a known and commonly measured clinical tool that could have improved mortality prediction.
2. Related to above. Levine et al. colleagues created the PhenoAge and BioAge phenotypes in UK Biobank (<https://doi.org/10.1111/ace.13376>). Arguably her BioAge is a more relevant comparison to the results presented here. Why is only PhenoAge included as a comparison? Justify not comparing to BioAge in the Discussion (or if easy, also include BioAge comparison).
3. Figure 1a is a great visualisation as it has 2 purposes early in the paper: list the biomarkers used, and rank by importance for the BA prediction. However, I do not think bar charts are the best way to present the C-index results. Figures 1b, 1c and 2 would be better presented as a forest plot, in my view, with the axis substantially truncated (e.g., ranging from 0.65 to 0.8 to better show the detail) and estimates + CIs values included (it is currently not possible to see whether the 95% CIs overlap between the C-index values of the different models).
4. Figure 4 presents results of the BA estimation associations with mortality, and made me realise it is not clear to “casual” readers what is meant by the “test set”. Please can the authors be more explicit in the figure legend that this is in a separate validation sample that was not used in the model training, and include number of participants (is it 20% of the 307,000? Where the other 80% was used for training?). A flow chart of the study design may help. Figure 4 b-d could have the y-axis truncated for clarity, and use exact p-values.
5. In the Discussion you comment on the similarity of the ENC results with those from PhenoAge, but in Figure 3 it looks like the creatinine coefficient is actually in the opposite direction. If I am reading this correctly (if not, clarify the figure?) then could you comment on this difference? Specifically, why *lower* creatinine (coefficient <0) would be correlated with increased biological age. I note that sensible sensitivity analyses were performed (excluding cystatin c and then creatinine, in case something was happening here) and actually creatinine is not significant when cystatin c is removed. Given this is the only major difference to the Levine PhenoAge it is worthy of Discussion.

Biological Age Estimation Using Circulating Blood Biomarkers

23 August 2023

Response to Reviewers of our Submission to Communications Biology

Please note that all references to line numbers in the text refer to the line numbers in the revised “clean” version of the manuscript, rather than line numbers in the marked-up version.

Reviewer 1

r1.1 Overall comment: However, the manuscript lacks a thoughtful consideration of what biological age is and the measurement approach it pursues is therefore somewhat challenging to understand. Moreover, the claims made don’t match the level of validation undertaken. This is important work. But the paper needs more. Suggestions are below.

We agree and have addressed these comments as set out below.

r1.2 BA should be defined the first time it is introduced in the introduction. The initialization needs to be spelled out. But also the authors need to explain what they think biological aging means and what a measure of it should represent. Then then they need to explain why they think multivariate hazard regression of mortality on blood chemistry parameters is a good way to develop a measurement of that construct.

We agree and have now added a definition as the first paragraph of the Introduction (lines 45 - 51). We have also inserted a sentence in lines 84 - 86 clarifying the link between mortality risk and BA.

r1.3 The limitations of the UKB setting should be articulated – at least in the discussion, but perhaps also up front. The PhenoAge to which they compare their measure was derived in a representative sample of the US adult population. UKB is not that. How might the restricted age range, high-SES skew, and other features of the UKB sample affect BA algorithm development?

We agree and have highlighted such limitations in our Discussion. In addition, in this revised version, we have added a SES stratified analysis in lines 161 -171 and Fig 1. We now address the question of applicability of our algorithm on various age ranges (lines 304-311, Fig 4), healthy vs sick sub-populations (lines 148-159, Fig 1) and socio-economic status (lines 161-171, Fig 1) and in the discussion from line 363).

r1.4 It appears that all validation testing was conducted using the same UKB data with which the algorithm was trained. This is not sufficient to demonstrate anything other than the potential value of an algorithm and should not be used as the basis for any comparison to PhenoAge.

We have now performed training only in England and Wales. All validation testing (for models developed in our analysis, all stratifications, the sex and age only model and PhenoAge) is now performed on the geographically distinct Scottish data set, which was not used in the development of any models. This point is now explained in lines 77 - 80. We have added wording in the Discussion recognising the geographic distance between PhenoAge training and Scottish UKBB validation sets (line 374-379).

r1.5 PhenoAge was developed in the NHANES III sample and validated in the Liu et al. article in fully-independent data from NHANES IV. At minimum, the new algorithm should be trained and tested in distinct subsets of the UKB sample. But would be much better to test the algorithm in a fully independent dataset.

See prior comment

r1.6 In order to compare performance of this new measure to PhenoAge, the authors should perform their analysis in an independent dataset (i.e. one not used to train either measure). It is not clear if this was done from the paper. The NHANES IV data analyzed by Liu et al. are publicly available and would provide one setting for this, although the difference in nation setting could bias results toward PhenoAge.

Yes, we agree in principle with this suggestion. However, in practice, NHANES does not have the assays we need to perform the head-to-head analysis. We believe, as suggested by the reviewer, that our Scotland vs England approach is a pragmatic compromise that achieves the underlying objective. We have, however, tightened wording at lines 374 -379, to reflect the relative proximity from Scotland of NHANES versus English/Welsh UKBB.

r1.7 The characterization of the improvement relative to PhenoAge should be specified both in relative % increase and absolute percentage point increase terms. 10% better is fine. But this reflects a 2-percentage-point improvement in discrimination. In the abstract and introduction, the authors should at least report the C-statistic for their new predictor. (it's actually quite striking that an order-of-magnitude + increase in sample size for training yields such modest improvement in the C-statistic: 0.74 for PhenoAge vs. 0.76 for this model)

Yes, agreed. We have added the absolute values alongside the relative performance improvement in the Abstract.

Reviewer 2

Major:

r2.1 The target outcome is not age but mortality, but the authors use the term Biological Age. Even for the Biological Age Acceleration (BAA), this index indicates the difference in mortality (not age) between CAGE-based information vs. blood-based information (which is not necessary equivalent to the general BAge, the age estimate based on biological information). This terminology should be clarified. It is parallel to PhenoAge, but not to other studies that predict age itself based on omics (see below).

We agree and have now added a definition of BA as the first paragraph of the Introduction (lines 45 - 51). We have also inserted a sentence in lines 84 - 86 clarifying the link between mortality risk and BA.

r2.2 The meaning of comparison with “Impute & Full Cox Model” (in Fig. 2) is unclear. For the analysis purpose, the authors first set the variables that is not included in a target panel to NAs for all people in the testing set (regardless these values were actually detected in the original data), and then imputed these NAs based on the detected real values of the people in the training set. Practically (as the authors claimed), this imputation procedure is impossible. So, I don’t understand why they presented “Impute & Full Cox Model” in Fig. 2. In other words, the comparisons between the Full ENC model (blue dashed line) and each “Bespoke Cox Model” (red bars) seem sufficient to claim their conclusion. If this is a misunderstanding, the authors should clarify.

We apologize for any confusion here. We agree it is impossible for an investigator merely in possession of our model coefficients and a subject's (partial) set of biomarker results. As the imputation model is also needed (in practice a subset of the test data, or similar dataset with all the assays). However, for investigators with such a representative dataset/similar dataset our procedure is practical. It is important to note there are more than 33 million possible subsets for our 25 markers - it is impractical to build all those models in advance and less practical to build the Full Elastic Net Model based on the specific set of markers available for a subject, from scratch, each time a new set of biomarkers is provided.

To clarify this, we have amended the text at line 335. The Impute and Full Cox model is the approach we have implemented successfully at Humanity (recognising it requires us to be in possession of, in our case, 10,000 test subjects with full data, as well the current subject with a restricted number of assay results).

r2.3 From the health-stratified analysis (Fig. 1c), the authors concluded that their Full ENC model work similarly for both Healthy and Sick (lines 130–131). However, the reason why they chose to focus on self-reported condition is not clearly mentioned. Based on Fig. 3a, age was the largest predictor for mortality. So, rather than (or in addition to) health status, the authors could perform the stratified analysis with age. In this age-stratified analysis, the model may not work similarly across age-stratified groups, but it would be a rather interesting point; e.g., they could check cystatin c and creatinine while considering their analysis about them (lines 225–231).

We appreciate this comment. We used self-reported status due to simplicity and its availability at assessment date (line 150). We have stratified by age, using 3 age groups in lines 304 onwards, as well as illustrated in Fig 4 as per the methodology used in the age group validation of PhenoAge. The stratification by health status was done in addition to this, to test whether our model was merely identifying sick individuals based on blood biochemistry measures. The stratification allowed for an assessment of model performance when controlling for health status.

r2.4 Similar studies have been performed on the same db albeit with slightly different algorithms/outcomes, e.g. <https://pubmed.ncbi.nlm.nih.gov/33693684/> or <https://www.medrxiv.org/content/10.1101/2022.10.05.22280735v1>, in addition to Levine et al. The authors should better stress their novelty beyond a slight improvement in performance of the model itself.

Thank you, we missed these studies during our literature review. We accept we should have mentioned these works. This is now done at lines 87, 221, and 225.

r2.5 The authors claim that their BA models demonstrated a 9.2% relative increase in predictive value when compared against PhenoAge. The authors should discuss what component of their methods causes this (modest) increase in performance.

We have added "The improved model performance likely arises from the large volume of data used for training, and, more importantly, the additional biomarkers cystatin C and red blood cell distribution width being available." to the discussion, line 324.

r2.6 The ENC model was both trained and evaluated on UKBB data, as the authors acknowledge. To improve the generalizability of their results and substantiate their claim for improved performance, the authors should apply it for the NHANES dataset as an external validation of not only the model performance itself, but also the downstream analysis of feature inference.

We agree this could be desirable, but not practicable at this stage, due to the amount of work involved and the absence of cystatin C (amongst others) in NHANES. We believe, as suggested by another reviewer, that our Scotland vs England approach is a pragmatic compromise that achieves the underlying objective. We feel by using a geographically distinct set, we have followed the spirit of this comment, and note that Liu et al. used the same geographical range, but a distinct period in time in their validation. We feel both approaches are broadly equally reasonable. We have included text in 374 to recognise the geographic proximity of validation. We have also compared the PhenoAge model's features to ours (where UKBB had a broader set of variables from which to select) and point out that all the markers in the PhenoAge model were also stably selected under our approach (lines 239 -249 and Fig 3b)

Minor:

r2.7 Sex stratified BA models were shown to have significantly different predictive features (e.g. Earls et. al. The Journals of Gerontology: Series A, 2019). The authors do perform stratified model, but limit the discussion to overall performance, rather than focusing on the sex-specific features of both the BA model and the delta age analysis.

We have expanded on the sex-specific nature of stably selected features by reporting their coefficients and selection proportions in the Supplementary Notes file (see table SN4). We report the coefficients for stably selected features in males-only and female-only groups and comment on the noteworthy differences.

r2.8 Some key papers are not cited (mentioned above)

Response as per comment r2.4

Reviewer 3

r3.1 Overall Comment: I find the paper to be well-written, the analysis thorough, and the results interesting. In particular, I am glad to not see another “biological age” paper where the model is trained to predict chronological age! Instead the authors use the far more interesting and relevant phenotype for prediction/intervention: “mortality.”

Overall, I have no concerns, however it is disappointing that the study authors (i.e., “Humanity Inc.”) have not released the BA model weights despite drawing heavily from previously published (and openly accessible) efforts (especially the Levine 2018 paper, which uses similar methods and this work extends to UKB), meaning their results cannot be used by others. This massively limits the impact and general interest in the work.”

Model weights can be found in Supplementary Table 4, alongside the PhenoAge coefficients. We have also created a public GitHub repo, with R code, which directly guides readers on how to perform our BAA calculation.

r3.2 1. Though telomeres are mentioned in sentence 1 as previously used BA markers they are not included in this study. Why not? I could see the argument that it is not commonly measured etc, so please justify the choice of 57 markers more clearly, and the exclusion of some known ageing biomarkers e.g., telomere length. Similar question for blood pressure: why not include? I know not a “blood” biomarker per se, but a known and commonly measured clinical tool that could have improved mortality prediction

We accept this point which is essentially one of scope. We explain our omission of telomeres/CpG measures in lines 56, and note at line 346 the desirability of future work extending scope to these and other biomarkers.

r3.3 2. Related to above. Levine et al. created the PhenoAge and BioAge phenotypes in UK Biobank (<https://doi.org/10.1111/ajcl.13376>). Arguably her BioAge is a more relevant comparison to the results presented here. Why is only PhenoAge included as a comparison? Justify not comparing to BioAge in the Discussion (or if easy, also include BioAge comparison).

This is a fair point, but the extension is beyond the scope of our present work. We also note that comparison of the outcome based (mortality) PhenoAge makes a degree of sense for our outcome based (mortality) clock, and, as with our analysis, PhenoAge limited to the feature scope to blood biomarkers, whilst BioAge made use of other measures such as pulse and blood pressure. We also wonder if Levine considered PhenoAge a better measure of aging, leading to its use as the basis of the epigenetic clock. We have added commentary in line 449 onwards: "Our models were compared against the performance of the well-known PhenoAge model. Kuo et al. (38) recently highlighted differing genetic dimensions of aging clocks, comparing PhenoAge and the alternative BioAge (39). We restricted the scope of our comparative analyses to the former as it was trained using blood biomarkers as predictors and mortality as outcome, as with our approach."

r3.4 3. Figure 1a is a great visualisation as it has 2 purposes early in the paper: list the biomarkers used, and rank by importance for the BA prediction. However, I do not think bar charts are the best way to present the C-index results. Figures 1b, 1c and 2 would be better presented as a forest plot, in my view, with the axis substantially truncated (e.g., ranging from 0.65 to 0.8 to better show the detail) and estimates + CIs values included (it is currently not possible to see whether the 95% CIs overlap between the C-index values of the different models).

Thank you for the suggestion. We agree and have amended the charts accordingly.

r3.5 4. Figure 4 presents results of the BA estimation associations with mortality, and made me realise it is not clear to “casual” readers what is meant by the “test set”. Please can the authors be more explicit in the figure legend that this is in a separate validation sample that was not used in the model training, and include number of participants (is it 20% of the 307,000? Where the other 80% was used for training?). A flow chart of the study design may help. Figure 4 b-d could have the y-axis truncated for clarity, and use exact p-values.

Thank you for the comment. Our test set is now made up of Scottish participants. We have made reference to this now in multiple locations in the revised version.

r3.6 5. In the Discussion you comment on the similarity of the ENC results with those from PhenoAge, but in Figure 3 it looks like the creatinine coefficient is actually in the opposite direction. If I am reading this correctly (if not, clarify the figure?) then could you comment on this difference? Specifically, why *lower* creatinine (coefficient <0) would be correlated with increased biological age. I note that sensible sensitivity analyses were performed (excluding cystatin c and then creatinine, in case something was happening here) and actually creatinine is not significant when cystatin c is removed. Given this is the only major difference to the Levine PhenoAge it is worthy of Discussion.

Yes, this is an interesting question, which we believe is address by adding the following text in line 350: "In the Full ENC model creatinine had a small, negative effect size, in contrast to a small, positive effect in the PhenoAge (14) model. However, we do not suggest that increased creatinine is a marker of good health (whilst Levine et al. (14) suggested it marks poor health), and, in fact, creatinine's coefficient was not statistically significant when cystatin C was not present. The contradiction of results against Levine et al. (14) is explained by the presence of cystatin C in our panel, which covaries with creatinine. We believe our results suggest increased creatinine in the presence of unchanged cystatin C may be desirable. In practice, increased creatinine will often be accompanied by increased cystatin C and a predicted worsening in health outlook."

Additional Change made by Authors.

r4.0 When we were preparing the manuscript, we realised that our Biological Age Acceleration term could be broken down more elegantly, by considering the null model (age and sex only) first and defining the age term based on this alone, making an identical but opposite adjustment in the term for biological age acceleration (BAA). These cancelling adjustments have absolutely no effect on the mortality hazard predictor, and thus our results (e.g., C-indices), but creates a BAA which is no longer intrinsically age dependent.

We have made these changes in our calculation of BAA. The process is described at 489 and the display items affected are only Fig4, and Fig SN5 and SN6 in the Supplementary Materials. An additional illustration is seen in lines 68 - 74 in the Supplementary Note, as well as in the R code file. Principally the spread of BAA is reduced somewhat, as there is no longer an intrinsic effect of chronological age on biological age acceleration (and the direct effect of chronological age on BA is 1:1).

REVIEWERS' COMMENTS:

Reviewer #1 (Remarks to the Author):

The authors have mostly addressed my comments. The new analysis separating out training in England/Wales and testing in Scotland is an important improvement.

My remaining comment is that I think the authors need to work a little harder at defining BA. It is fair to take the empirical approach of mortality risk prediction as the standard for BA estimation – this is what was done with the PhenoAge. But more needs to be said about what this is supposed to reflect and why the authors believe that it does so.

R1.2: the definition of BA needs more than “an individual’s functional age, represented by mortality risk.” What is functional age? How does that relate to mortality risk? There are many definitions of biological age out there. It’s ok to borrow one. But spell it out.

Reviewer #2 (Remarks to the Author):

I am satisfied with the authors' response. I have no further concerns or comments.

Reviewer #3 (Remarks to the Author):

Many thanks for responding to my comments. I apologise for missing your Supplementary Table 4 in my first review. I have no further comments and look forward to seeing the paper published.